# Integrated Omics Analysis Reveals Key Pathways in Cotton Defense against Mirid Bug (*Adelphocoris suturalis* Jakovlev) Feeding

**DOI:** 10.3390/insects15040254

**Published:** 2024-04-08

**Authors:** Hui Lu, Shuaichao Zheng, Chao Ma, Xueke Gao, Jichao Ji, Junyu Luo, Hongxia Hua, Jinjie Cui

**Affiliations:** 1National Key Laboratory of Cotton Bio-Breeding and Integrated Utilization, Chinese Academy of Agricultural Sciences, No. 38, Huanghe Road, Anyang 455000, China; luhui-111@163.com (H.L.); hnnydxjc@163.com (J.J.); luojunyu1818@126.com (J.L.); 2Hubei Insect Resources Utilization and Sustainable Pest Management Key Laboratory, College of Plant, Science and Technology, Huazhong Agricultural University, Wuhan 430070, China; huahongxia@mail.hzau.edu.cn; 3Green Agricultural Products Safety and Warning Laboratory, Research Center of Soil Resource Comprehensive Utilization and Ecological Environment in Western Inner Mongolia, Hetao College, Bayannur 015000, China; 4Henan Institute of Science and Technology, College of Life Science, Hualan St. 90, Xinxiang 453003, China; yndxzsc@126.com; 5Anhui Provincial Center for Disease Control and Prevention, Hefei 230601, China; machaoyzu@163.com

**Keywords:** *Adelphocoris suturalis* Jakovlev, cotton plant defense, α-linolenic acid metabolism pathway, fructose and mannose biosynthesis pathway, key regulators

## Abstract

**Simple Summary:**

This study delves into the intricate defense mechanisms of cotton plants against the primary pest *Adelphocoris suturalis* Jakovlev, which has become a major concern in Bt-cotton fields. Through advanced analytical techniques, this study identified specific changes in the cotton plant’s metabolism and protein expression triggered by *A. suturalis* feeding. These findings revealed a notable increase in the expression of certain α-linolenic acid metabolism pathway-related proteases and a simultaneous decrease in fructose and mannose biosynthesis-related proteases. These molecular responses shed light on the complex interplay between cotton plants and *A. suturalis*. Therefore, this study not only deepens our understanding of plant–insect interactions but also provides valuable insights for developing innovative strategies to control this cotton pest, thereby offering potential solutions for sustainable cotton cultivation.

**Abstract:**

The recent dominance of *Adelphocoris suturalis* Jakovlev as the primary cotton field pest in Bt-cotton-cultivated areas has generated significant interest in cotton pest control research. This study addresses the limited understanding of cotton defense mechanisms triggered by *A. suturalis* feeding. Utilizing LC-QTOF-MS, we analyzed cotton metabolomic changes induced by *A. suturalis*, and identified 496 differential positive ions (374 upregulated, 122 downregulated) across 11 categories, such as terpenoids, alkaloids, phenylpropanoids, flavonoids, isoflavones, etc. Subsequent iTRAQ-LC-MS/MS analysis of the cotton proteome revealed 1569 differential proteins enriched in 35 metabolic pathways. Integrated metabolome and proteome analysis highlighted significant upregulation of 17 (89%) proteases in the α-linolenic acid (ALA) metabolism pathway, concomitant with a significant increase in 14 (88%) associated metabolites. Conversely, 19 (73%) proteases in the fructose and mannose biosynthesis pathway were downregulated, with 7 (27%) upregulated proteases corresponding to the downregulation of 8 pathway-associated metabolites. Expression analysis of key regulators in the ALA pathway, including allene oxidase synthase (AOS), phospholipase A (PLA), allene oxidative cyclase (AOC), and 12-oxophytodienoate reductase3 (OPR3), demonstrated significant responses to *A. suturalis* feeding. Finally, this study pioneers the exploration of molecular mechanisms in the plant–insect relationship, thereby offering insights into potential novel control strategies against this cotton pest.

## 1. Introduction

Higher plants successfully colonize herbivore-rich environments by developing complex anti-herbivore feeding mechanisms [1]. When plants detect physical and chemical cues from insects, such as oral secretions and oviposition secretions, they can modify their proteins and metabolites [2]. Plants primarily resist herbivorous insects through jasmonate (JA), ethylene, and salicylic acid [3,4]. Specifically, the JA signaling pathway is crucial for this plant resistance [5,6]. Jasmonates (JAs), belonging to the oxylipin family, are key players in plant defense [7]. Initiation with α-linolenic acid (ALA) in the JA biosynthesis pathway produces key anabolic products like JA and its derivatives (e.g., methyl jasmonate, isojasmonic acid, methyl isojasmonate, etc.). In the ALA metabolism pathway, allene oxide synthase (AOS), allene oxide cyclase (AOC), and 12-oxophytodienoate reductase3 (OPR3) are crucial for JA synthesis [8,9,10]. ALA released as a stress signal through lipase activity on chloroplast membranes serves as the substrate for various oxidized lipids, including methyl jasmonate (MeJA) and other JA metabolites [11]. Increasing evidence suggests that the JA signaling pathway is crucial in plant defense, contributing to rice resistance against the leaf folder (*Cnaphalocrocis medinalis*) [12]. JA also mediates cotton plant resistance induced by *Helicoverpa armigera* and affects the growth and development of *Aphis gossypii* [13]. When rice plants are attacked by the brown plant hopper (*Nilaparvata lugens*), JA activates rice defense responses and gibberellin catabolism [14]. Phospholipase A (PLA), AOS, AOC, and OPR3, the key enzymes in the JA biosynthesis pathway, have been well reported in enhancing plant defense [9,14,15,16,17]. AOS and its family play a vital role in determining the resistance of rice to chewing and phloem-feeding herbivores [14,17].

Cotton (*Gossypium hirsutum L.*), a significant industrial crop worth 50 billion USD globally [18], has seen widespread adoption of transgenic Bt (*Bacillus thuringiensis*) cotton, reducing the need for broad-spectrum insecticides in China. However, this shift has resulted in frequent mirid outbreaks [19,20,21]. *Adelphocoris suturalis*, a mirid bug, has emerged as a top pest in the primary cotton-growing area. Both nymphs and adults of *A. suturalis* feed on cotton buds, tender shoots, leading to abscission, wilting, abnormal growth, and losses in lint production and quality [22]. These mirid bugs may eventually threaten the future of Bt cotton. Biological control methods, including transgenic plants, entomopathogenic microbes or a combination of both, have shown promising efficacy in pest management [23,24,25,26]. Many studies have focused on *A. suturalis* ecology and physiology [27,28,29,30]. However, current research on how *A. suturalis* responds to the mechanism of resistance defense in cotton is still lacking. Therefore, study of key pathways and key regulators will enrich our knowledge about plant defense and can help develop novel strategies to control mirid bugs, and most importantly provide the theoretical basis for transgenic breeding techniques in cultivating cotton against insect pests.

Detection and quantification of metabolites are achieved with techniques like quadrupole time-of-flight mass spectrometry (LC-Q/TOF-MS)-based metabolomics [31,32,33], and for the detection and quantification of proteins, quantitative isobaric tag for relative and absolute quantitation–liquid chromatography–mass spectrometry/mass spectrometry (iTRAQ-LC–MS/MS) proteomics is used. This sensitive method facilitates high-throughput protein identification and quantification [34], with applications in insects and plants like silkworm [35], brown planthopper [36], pine beetle [37], locust [38], and cotton [39]. These techniques provide a comprehensive view of dynamic proteomic and metabolomic variations, thereby enabling the identification of crucial proteins and metabolites essential for plant defenses.

The *A. suturalis*–cotton interaction forms the scientific basis for this pest control. Our study delves into the molecular-level intricacies of the interaction mechanism between cotton and *A. suturalis*. The specific objectives of this study were to identify insect-resistant genes and leverage transgenic technology in cotton for effective *A. suturalis* control. Understanding cotton’s insect resistance mechanism post-*A. suturalis* feeding holds great significance. This research aims to improve the theoretical framework of integrated pest management for *A. suturalis*. However, limited reports exist on cotton protein and metabolite changes after *A. suturalis* feeding [27,28]. Therefore, in this study, we employed proteomic and metabolomic approaches to investigate cotton plant defenses in response to *A. suturalis* feeding, explored changes, and identified potential molecular mechanisms.

## 2. Materials and Methods

### 2.1. Plant Material and Insect Infestation

Potted cotton plants (*G. hirsutum* CCRI 49) were transplanted into soil pots (15 × 17 cm), then put it into environmentally controlled chambers (50 × 60 × 110 cm) and maintained under either a 14 h (long day) or 10 h (short day) light period at 25 ± 2 °C and 65–70% relative humidity till the plants reached the four-leaf stage for further testing [13]. Subsequently, each cotton plant was placed in a 30 cm squared cage, and nine adult *A. suturalis* (starved for 12 h before inoculation) were introduced onto the leaves. A preliminary experiment showed that a 24–48 h timeframe is crucial for medium-term resistant defense. After 48 h of insect infestation, all insects were promptly removed. Cotton samples were immediately flash-frozen in liquid nitrogen and stored at −80 °C for further processing.

### 2.2. Metabolomic Sequencing

The primary aim of metabolomic analysis is to identify statistically significant and biologically relevant metabolites from the extensive pool of detected metabolites, followed by elucidating associated metabolic processes. LC–MS-based targeted metabolomic analysis was carried out as per a previous study [40,41]. Progenesis QI (ver. 2.2) software analyzed the MS data for peak (mz) retention time and ion area. The normalized data were then entered into SIMCA-P V11.0 (Umetrics, Umeå, Sweden) for PCA and OPLS-DA analysis. Univariate analysis was performed using *t*-tests, and the *p*-values from the *t*-tests after false-discovery rate (FDR) correction then produced *q*-values [41,42]. The results were significant when the *q*-value was <0.05. OPLS-DA was used to investigate the metabolite pattern changes. The differential metabolites were then selected when VIP values from the OPLS-DA model exceeded 1. FC ≥ 1.2 or ≤0.8 was used to determine differential metabolite variance between groups [41,42,43,44]. The importance of differential metabolites is connected to the FC value and VIP score, and the larger the value, the higher the score and the more important the substance is. Each metabolite’s related pathways were also listed by searching the KEGG (http://www.genome.jp/KEGG/, accessed on 16 June 2017) pathway database, while the metabolite molecular formulae of the matched metabolites were further identified by isotopic distribution measurement. Metabolomic sequencing included ten experimental replicates for each treatment, with group 1 (M48C) indicating uninfested controls grown for 48 h and group 2 (M48E) representing plants infested with insects for 48 h.

### 2.3. iTRAQ Sequencing

iTRAQ sequencing was conducted at BGI (Shenzhen, China). Using a reported phenol extraction method, cotton plant total proteins were extracted [45,46,47]. Protein concentrations were measured using the Bradford method [48]. The experiment comprised three independent biological replicates, each utilizing 25 μg total protein. Sequencing-grade trypsin (Promega) was used to digest the proteins at a 1:10 (w:w) ratio for 12 h at 37 °C. Subsequently, the samples were labeled using iTRAQ 8-plex kits (AB Sciex Inc., Foster, CA, USA) according to the manufacturer’s instructions. The samples were labeled with P48E (insect feeding, 48 h treatment) and P48C (control without insect feeding, 48 h). LC-MS/MS analysis was carried out as described previously [45,49], on an AB SCIEX Triple TOF 5600 system (Foster, CA, USA) with a Nanospray III source (AB SCIEX, Concord, ON, USA) featuring a pulled quartz tip as the emitter (New Objectives, Woburn, MA, USA). The MS operated with an RP ≥ 30,000 FWHM for TOF MS scans. Mascot version 2.3.02 search engine by Matrix Science, Boston, MA, USA, was used for protein identification and quantitative analysis, as previously reported [50,51,52]. The ExPASy website provided by the Expert Protein Analysis System was utilized for protein sequence prediction and retrieval, as reported previously [53]. Differentially upregulated and downregulated proteins were considered significant if their FC values were ≥1.2 or ≤0.8; *q*-values obtained through FDR correction of *p*-values indicated statistical significance at *q* < 0.05 [39,45,46]. Blast2GO was used to annotate protein functions against the non-redundant NR protein database on NCBI. The KEGG (http://www.genome.jp/KEGG/, accessed on 16 June 2017) and GO (http://geneontology.org/, accessed on 5 July 2017) databases were used for categorizing and grouping identified proteins. For proteomic sequencing, each treatment was performed in triplicate.

### 2.4. Quantitative Real-Time PCR (qRT-PCR)

For qRT-PCR analysis, six proteins were selected. These proteins included allene oxidase synthase (AOS), phospholipase A (PLA), alkene oxidative cyclase (AOC), alcohol dehydrogenase (ADH), and two 12-oxophytodienoate reductase 3 (OPR3-1 and OPR3-2), and experiments were conducted corresponding to 3 h, 6 h, 12 h, 24 h, 48 h, and 72 h of insect infestation, respectively. Control plants not subjected to infestation, were grown for the same duration. The leaves were promptly frozen in liquid nitrogen and stored at −80 °C until further use.

Total RNA was extracted using the RNAprep Pure Plant Kit (TIANGEN Biotech, BeiJing, China) following the manufacturer’s protocol [28]. First, 1 μg RNA was reverse-transcribed to cDNA via a PrimeScriptTM RT reagent kit (perfect real time) (TaKaRa, Dalian, China) as per the manufacturer’s protocol. The RT-qPCR, conducted on an Eppendorf Mastercycler eprealplex 2.2 (Eppendorf, Hamburg, Germany), utilized GoTaq Qpcr Master Mix (Promega, Madison, WI, USA) with three biological and three technical replicates. The thermal cycle conditions for qRT-PCR were 95 °C for 2 min, followed by 40 cycles of 95 °C for 15 s and 60 °C for 1 min. Relative gene expression was analyzed through the 2^−△△Ct^ method [54]. The mean Ct value of three technical replications in each sample was used for the gene expression analysis, and the relative expression level of each treatment was the average of the three biological replications, and incorporated two housekeeping genes (*GhHis3*; GenBank accession AF024716, and *Gossypium hirsutum ubiquitin* 7 (Gh*UB7*, GenBank accession DQ116441.1) according to the literature [55]. The three-hour control group served as the reference sample for data normalization. All qRT-PCR primer pairs were designed using the online Primer Quest Tool (http://sg.idtdna.com/PrimerQuest/Home/Index, accessed on 12 August 2017) and are detailed in Appendix A. Significance in expression level differences was determined by one-way ANOVA, and means were separated using Tukey’s HSD (IBM SPSS Statistics v 22.0).

### 2.5. Data Analysis

Data analysis and chart preparation were conducted using Microsoft Excel 15.37, Mac Preview (8.1), Adobe Photoshop (2020), and GraphPad Prism 6. Data were analyzed using IBM SPSS Statistics 22.

## 3. Results

### 3.1. Metabolite Identification

#### 3.1.1. Identification of Differential Metabolites

A comprehensive metabolomic analysis was conducted on cotton plants infested by *A. suturalis* for 48 h. The initial mass spectrometry data comprised 7790 positive ions and 4325 negative ions. Subsequent PCA analysis of quality control (QC) samples revealed that PCA1 contributed 32.32% and PCA2 contributed 16.78%, suggesting their suitability for PCA. Importantly, QC samples exhibited clustering, affirming instrument stability and data reliability (Appendix A). In this study, both positive ion (ESI+) and negative ion (ESI-) modes collectively contributed over 50% (ESI+ = 57.08%, ESI− = 52.14%). This indicates that these principal components effectively represent and interpret the significant metabolite changes induced by *A. suturalis* feeding on cotton. Furthermore, PLS-DA analysis was employed to maximally highlight differences among test groups. The ESI+ model parameters were R^2^ = 0.86, Q^2^ = 0.62, while ESI− model parameters were R^2^ = 0.90, Q^2^ = 0.69. The minimal difference between R^2^ and Q^2^ values (0.2–0.3) suggests the reliability and effectiveness of the current PLS-DA model [56] (Appendix A). Under this standard (VIP ≥ 1, FC ≥ 1.2 or ≤0.8, and *q* < 0.05), there were 496 differential positive ions (122 downregulated, 374 upregulated) and 53 differential negative ions (7 downregulated, 46 upregulated) (Appendix A). A total of 496 cationic metabolites categorized under 11 types were detected in the cotton metabolome (Figure 1A, Appendix A), including terpenoids, alkaloids, phenylpropanoids, flavonoids or isoflavones, organic acids and their derivatives, phenols and their derivatives, carbohydrates and carbohydrate conjugates, lipids and lipid molecules, amino acid peptides and their analogues, nucleotides and their analogues, etc. These results suggested that metabolites were upregulated, which may induce cotton’s defense against herbivorous insects. All differential cationic metabolites (496) and additional analytical details are listed in the Appendix A.

Our results showed that the cotton plant responded to the feeding stress of *A. suturalis* by increasing its secondary metabolites. These metabolites, including terpenes, alkaloids, phenylpropanoids, and flavonoids, were most significantly upregulated post-*A. suturalis* feeding (Figure 1B, Appendix A). The terpenes germacrene D (*q* = 0.043) and alpha-barbatene (*q* = 0.043) showed the highest upregulation, with FC up to 46.8 and VIP up to 3.6. Among alkaloid metabolites, betaine aldehyde (*q* = 0.043) and acetylcholine (*q* = 0.036) were upregulated, reaching FCs of 46.8 and 29.4, and VIP scores of 3.6 and 4.0, respectively. The phenylpropanoid metabolites, including coumarin (*q* = 0.002) and cinnamaldehyde (*q* = 0.020). showed overall upregulation, reaching FCs of 5.8 and 5.3 and VIP scores of 3.2 and 2.4, respectively. All secondary flavonoids were significantly upregulated, with ononin (*q* = 0.043) and syringetin (*q* = 0.0002) exhibiting FCs up to 5.5 and 5.4 and VIP scores up to 2.3 and 3.3, respectively (Figure 1B,C, Appendix A).

#### 3.1.2. KEGG Annotation and Functional Analysis of Differential Metabolites

The KEGG database was used to annotate 496 differential metabolites into 65 pathways after cotton consumption compared to the control group. The pathways included the sesquiterpene and triterpene biosynthetic pathway (map00909), arachidonic acid metabolic pathway (map00590), tryptophan metabolic pathway (map00380), etc. (Table 1 and Appendix A).

#### 3.1.3. The Two Key Metabolic Pathways

Plant resistance is multifaceted. For example, plants can reduce their own nutrients to affect the growth and development of insects, achieving insect resistance [57]. In addition, α-linolenic acid acts as a stress signal released through lipase activity on chloroplast membranes and serves as the substrate for various oxidized lipids, including methyl jasmonate (MeJA) and other JA metabolites [7]. Therefore, we chose to comprehensively analyze two pathways of interest: the fructose and mannose metabolism pathway and the ALA metabolism pathway. The fructose and mannose metabolism pathway showed six upregulated metabolites (*q* = 0.028) and eight downregulated metabolites (*q* = 0.032) (Figure 2A,B), while the ALA metabolism pathway exhibited fourteen upregulated metabolites (*q* < 0.047) and two downregulated metabolites (*q* < 0.045) (Figure 2C,D, Appendix A).

### 3.2. iTRAQ Identification of Proteins

Cotton proteomic analysis using iTRAQ involved three replicates, as previously described [39,45,47]. A total of 371,575 and 374,246, spectra were obtained by P48E treatment and P48C control, including 52,750 and 48,103 unique spectra and 17,865 and 16,116 peptide segments, respectively. Furthermore, 12,906 and 11,903 unique peptide proteins were obtained, and 5520 and 5210 unique peptide proteins were identified in P48E and P48C, respectively. Overall, a total of 8302 proteins were identified from the three samples (Appendix A).

#### 3.2.1. Functional Categories of Differentially Expressed Proteins

Of all 8302 proteins, there are a total of 1252 differentially expressed proteins (*q* < 0.05, 775 upregulated and 477 downregulated, Appendix A) in the P48E vs. P48C control comparison (Figure 3A). These findings indicate that plants adjust protein levels in response to *A. suturalis* attacks, activating plant defense responses. GO enrichment analysis of differentially expressed proteins (DEPs) for P48E and its P48C control showed them being classified into three groups: biological process, cellular component, and molecular function (Figure 3B, Appendix A).

#### 3.2.2. Pathway Enrichment of Differential Proteins

Furthermore, the metabolic pathway enrichment analysis of cotton post-feeding showed the sequencing results for feeding treatment (P48E) and control (P48C). The analysis identified 1569 differential proteins enriched in 35 metabolic pathways (*q* < 0.05) (Appendix A), with the top 20 pathways depicted in Figure 4A. Subsequent examination indicated a higher number of upregulated proteins in defense-related metabolic pathways compared to downregulated ones (Figure 4B), exemplified by the phenylpropanoid biosynthesis pathway (30 upregulated proteins, 15 downregulated), ALA metabolism pathway (17 upregulated and 2 downregulated), and fructose and mannose metabolism pathways (7 upregulated and 19 downregulated).

### 3.3. Integrative Metabolomic–Proteomic Analysis

The proteins and corresponding metabolites were considered to be correlated if they were both expressed at the same metabolic pathway. Integrated analysis of the proteome and metabolome revealed significant changes in metabolic pathways in cotton after *A. suturalis* feeding, including phenylpropanoid biosynthesis, amino sugar and nucleotide sugar metabolism, photosynthesis pathway, fructose and mannose metabolism pathway, glutathione metabolic pathway, pentose phosphate pathway, alpha-linolenic acid metabolism, etc. (Figure 5). Therefore, fructose and mannose metabolism, as well as ALA metabolism, were selected for further analysis.

#### 3.3.1. Integrating Proteomics and Metabolomics for Fructose and Mannose Metabolism Pathway Analysis

In the integrative proteo-metabolomic analysis of fructose and mannose metabolic pathways, we identified 7 upregulated and 19 downregulated proteases (Figure 6, Appendix A). The upregulated proteases include phosphofructokinase (*q* = 0.001), epidermis-specific secreted glycoprotein EP1 (*q* = 0.038), xylose isomerase (*q* = 0.001), idonate-5-dehydrogenase (*q* = 0.002), and fructose-bisphosphate aldolase3 (*q* = 0.001). The downregulated proteases comprise fructose-bisphosphate aldolase (*q* = 0.043), triosephosphate isomerase (*q* = 0.001), fructokinase (*q* = 0.001), endo-1,4-beta-mannanase (*q* = 0.017), hexokinase (*q* = 0.001), triosephosphate isomerase (*q* = 0.001), and fructokinase-1-6-bisphosphatase (*q* = 0.001), etc. These differential proteases reached significant levels (*q* < 0.05), with some at extremely significant levels (*q* < 0.01).

The metabolomic data indicated six upregulated and eight downregulated metabolites (Figure 6, Appendix A). The upregulated metabolites include D-allose (*q* = 0.028), D-mannose (*q* = 0.028), L-sorbose (*q* = 0.028), L-rhamnonate (*q* = 0.028), L-fuconate (*q* = 0.028), and D-fructose (*q* = 0.028). Downregulated metabolites comprised D-allose 6-phosphate (*q* = 0.032), D-allulose 6-phosphate (*q* = 0.032), D-mannose 1-phosphate (*q* = 0.032), D-mannose 6-phosphate (*q* = 0.032), D-fructose 1-phosphate (*q* = 0.032), beta-D-fructose 2-phosphate (*q* = 0.032), sorbose-1-phosphate (*q* = 0.032), and GDP-4-dehydro-6-deoxy-D-mannose (*q* = 0.024) (Figure 2 and Figure 6). These differential metabolites reached significant levels (*q* < 0.05), with some being extremely significant (*q* < 0.01).

The observed changes in both proteins (seven upregulated and nineteen downregulated) and metabolites (six upregulated and eight downregulated) strongly suggest that plants employ a defense strategy against insect predation by reducing their own nutrient consumption. This subsequently affects the growth and development of *A. suturalis*, and this is consistent with what was observed during the experiment (e.g., development process is prolonged).

#### 3.3.2. Integrating Proteomics and Metabolomics for ALA Metabolism Pathway Analysis

In the integrative metabolomic–proteomic analysis of the ALA metabolism pathway, 17 upregulated and 2 downregulated proteases were identified (Figure 7, Appendix A). The upregulated proteases included salicylate O-methyltransferase (*q* = 0.032), AOS (*q* = 0.021–0.038), 3-ketoacyl-CoA thiolase 2 (KAT2; *q* = 0.040), phospholipase A1-Iigamma (PLA; *q* = 0.001), 4-coumarate--CoA ligase-like 5 (*q* = 0.001), acyl-coenzyme A oxidase 2 (ACX2; *q* = 0.003), acyl-coenzyme A oxidase 4 (ACX4; *q* = 0.011), OPR3 (*q* = 0.0110), alkene oxidative cyclase 4 (AOC; *q* = 0.001), et al. The downregulated proteases included alcohol dehydrogenase class-P (ADH; *q* = 0.001), and phospholipase A1-IIdelta (PLA; *q* = 0.011). These differential proteases reached significant levels (*q* < 0.05), with some at extremely significant levels (*q* < 0.01).

Metabolomic data for the ALA metabolism pathway showed 14 upregulated and 2 downregulated metabolites (Figure 7, Appendix A). The upregulated metabolites included 3-hexenal (*q* = 0.029); 12,13-EOT (*q* = 0.022), 9,10-EOT (*q* = 0.022), colnelenic acid (*q* = 0.022), etherolenic acid (*q* = 0.022), 10-OPDA (*q* = 0.022), 12-OPDA (*q* = 0.022), JA (*q* = 0.018), 7-isojasmonic acid (*q* = 0.018), stearidonic acid (*q* = 0.047), 3,6-nonadienal (*q* = 0.041), 9-oxononanoic acid (*q* = 0.038), MeJA (*q* = 0.032), and 7-isomethyljasmonate (*q* = 0.032). The downregulated metabolites were 3-hexenol (*q* = 0.045) and volicitin (*q* = 0.031). These differential metabolites reached significant levels (*q* < 0.05).

Based on the changes seen in both proteins (17 upregulated and 2 downregulated) or metabolites (14 upregulated and 2 downregulated), we can conclude that plants resist insect predation by increasing the production of defense regulators like proteins and metabolites to defend against *A. suturalis* feeding.

### 3.4. Analysis of Key ALA Metabolism Pathway Regulators

Upon analysis, significant alterations (upregulated) in the ALA metabolism pathway were observed in both proteomic and metabolomic analyses. The integrated proteomic and metabolomic analysis revealed predominantly downregulated metabolites, except for volicitin and 3-hexenol, whereas most related proteins were upregulated. These proteins included AOS, PLA, AOC, and OPR3, etc. The outcome suggests an increase in upstream regulator proteins, leading to a decrease in downstream metabolites within the ALA metabolism pathway. This protein and metabolite reconfiguration appears to be a plant response to insect-feeding stress, indicating the potential significance of the ALA pathway in cotton defense against herbivores. Key regulatory proteins included AOS, PLA, AOC, and OPR3.

To verify that protease regulators respond to mirid bug feeding, further investigation focused on six selected proteases from the ALA metabolic pathway, comprising four upregulated proteins (Appendix A), i.e., AOS (CotAD_35840), PLA (CotAD_52791) and AOC (Cotton_D_gene_10007846), OPR3 (OPR3-1; CotAD_59461, and OPR3-2; Cotton_D_gene_10037325) and one downregulated protein, i.e., alcohol dehydrogenase ADH (CotAD_64340). These proteins underwent qRT-PCR analysis to confirm expression at the transcription level, validating the sequencing results (Figure 8).

The results showed that phospholipase A (*PLA*) gene expression increased progressively with feeding time, finally peaking at 48 h before decreasing significantly differently from the control group (Figure 8A, *p* < 0.01). Allene oxidase synthase (*AOS*) gene expression was highest at 12 h, showing a significant difference from the control group (Figure 8B, *p* < 0.01). Allene oxide cyclase (*AOC*) gene expression in cotton leaves gradually increased during *A. suturalis* feeding, peaking at 24 h, and then declined. *AOC* expression at 6 h, 12 h, 24 h, 48 h, and 72 h post-insect feeding significantly differed from the control group (Figure 8C, *p* < 0.01). These results suggest *AOC* has a relatively faster response in upper plant parts to *A. suturalis* feeding. *OPR3-1* and *OPR3-2* displayed similar expression levels, with both reaching significant differences (*p* < 0.05) at 12 h (*p* < 0.05) and 24 h (*p* < 0.01) compared to the control group (Figure 8D,E). ADH was downregulated, reaching significant differences at 3 h, 6 h and 24 h (Figure 8F, *p* < 0.01).

Through dynamic time variation analysis (Figure 8G), we conclude that *PLA* expression gradually increases, peaking at 48 h and subsequently decreasing. *AOC* demonstrated heightened sensitivity and rapid response to *A. suturalis* feeding, particularly. *OPR3-1* and *OPR3-2* maintained moderate levels (Figure 8G, pink and blue curves), while *AOS* experienced notable fluctuations (Figure 8G, green curve). This suggests that plants can adjust protein and gene levels to combat insect feeding. Understanding why plants modify proteins and metabolites post-insect attack is crucial for elucidating the plant–insect relationship, and our results provide the theoretical basis for it.

## 4. Discussion

Plants in natural environments encounter diverse herbivores, pathogens, and other biotic and abiotic stresses [58,59,60]. To survive these challenges, plants have evolved intricate mechanisms to respond to herbivore attacks, reshaping their transcriptomes, proteomes, and metabolomes upon detecting physical and chemical cues from herbivores [3,60,61]. These responses involve specific alterations in metabolism, gene expression, and plant growth and development patterns [1,62,63,64]. When attacked by herbivore feeding or egg laying, the injury initiates complex reactions that ultimately lead to plant defense [65], while insects can release effectors that disturb host plant defense responses for fitness, and those effectors are crucial components in insects and the host plant [66]. Plants can recognize herbivore-associated molecular patterns (HAMPs) and trigger various types of defense signal transduction, be it against chewing herbivores or piercing–sucking insects [67]. Effectors in oral secretion (OS) have been well reported to trigger or interfere with plant defense [68]. The first reported OS effector was fatty acid conjugates (FACs) of caterpillars [69]. Recently, researchers found an effector, HARP1, from *H. armigera* oral secretion. HARP1 may affect JA-responsive genes and make the plants more suitable for insect feeding [66]. In recent years, JA has attracted great attention and its functions in plant stress responses against pathogens, herbivorous as well as in development, have been well studied [70]. In our study, we identified five genes (*PLA, AOS, AOC, OPR3-1, OPR3-2*) enriched in the ALA signaling pathway contributing to increased JA levels (Figure 7). In addition, the ALA metabolism pathway demonstrated a pronounced response to *A. suturalis* plant feeding at both protein and metabolic levels. We also observed significantly increased expression of metabolites and proteases in the ALA pathway. This is consistent with a previous report observed in rice in response to stress induced by the rice stem borer [40]. Consequently, we hypothesized that the ALA metabolic pathway plays a crucial role in cotton response to mirid bugs, and the upregulated enzymes in the ALA metabolism pathway, including PLA, AOS, AOC and OPR3, positively contribute to JA biosynthesis (Figure 7). This finding aligns with previous studies indicating that AOS, AOC, and their families are vital in determining the resistance of rice to phloem-feeding herbivores [15,70]. However, our results also suggest that downregulated ADH in the ALA metabolism pathway negatively impacts JA biosynthesis (Figure 7).

Over the years, evidence has accumulated that plant primary and secondary metabolism profiles influence insect behavior [66,71]. Some plants may reduce nutrients that are required for insect survival and reproduction [57]. In addition, secondary metabolites may deter insects from feeding or egg laying and can also provide defensive functions and regulate defense signaling pathways against herbivores [65]. In our study, the integrated proteomic–metabolomic data analysis from two key metabolic pathways (fructose and mannose biosynthesis and ALA metabolism) indicated that cotton responds to feeding stress by decreasing nutrient content (e.g., fructose and mannose metabolic pathway) and increasing metabolite and protein content in related pathways (e.g., ALA metabolic pathway). These findings align with previous studies proposing two main hypotheses: active defense and nutrient stress [72,73]. The active defense hypothesis suggests that plants actively responded to insect feeding, resulting in increased metabolite content. In contrast, the nutrient stress hypothesis suggests that insect pests cause a reduction in plant nutrient content, affecting plant-eating insect growth and development [72,73]. Our results demonstrate that the cotton plant respond to *A. suturalis* feeding stress by increasing its secondary metabolites, including terpenes, alkaloids, phenylpropanoids, and flavonoids, which were most significantly upregulated post-feeding (Figure 1B,C). Correlational studies have identified small molecules with antifeedant or toxic effects on herbivorous insects. Notably, terpenoids, a metabolically diverse class of plant secondary metabolites, contribute to plant defenses [58,74]. Defensive properties of various compounds, such as furanocoumarins, cardenolides, tannins, saponins, glucosinolates, and cyanogenic glycosides, are well established [13,71].

In summary, the defense response of cotton to *A. suturalis* is a combined manifestation of both the nutrient stress hypothesis and the active defense hypothesis. This study offers a theoretical foundation and identifies candidate target genes for enhancing cotton plant defense resistance through molecular breeding. Therefore, these findings could pave the way for development of novel control strategies against this insect pest. Further studies on those regulators will enrich our knowledge about plant defense and help design novel strategies to control insect pests, most importantly helping us develop novel transgenic breeding techniques in cultivating cotton against insect pests.

## 5. Conclusions

In this study, we propose that *A. suturalis* triggers a complex cotton plant resistance process encompassing the upregulation of plant secondary metabolites (terpenes, alkaloid, phenylpropanoid, and flavonoids, Figure 9A), a reduction in nutrient content (metabolites and proteins of fructose and mannose biosynthesis pathway, Figure 9B), and an increase in the key pathway metabolites and proteins (such as those in the ALA metabolic pathway, Figure 9B). Our focus was on elucidating the signaling pathways for *A. suturalis*-induced plant resistance, particularly the ALA pathway, where most proteases are upregulated, leading to elevated JA levels. We propose that the ALA metabolic pathway significantly contributes to cotton’s defense response, with proteases like AOS, AOC, PLA, and OPR3 identified as key regulators in cotton resistance. Additionally, screening for anti-insect metabolites, proteins, or genes offers valuable insights for biological control of insect-associated pests.

Therefore, it is essential to acknowledge that insect-induced plant resistance is a multifaceted issue. This study specifically addresses cotton resistance under laboratory conditions. Further investigations are warranted, including assessing whether the gene expression increase persists under natural field conditions. Furthermore, long-term effects also merit further exploration.

## Figures and Tables

**Figure 1 insects-15-00254-f001:**
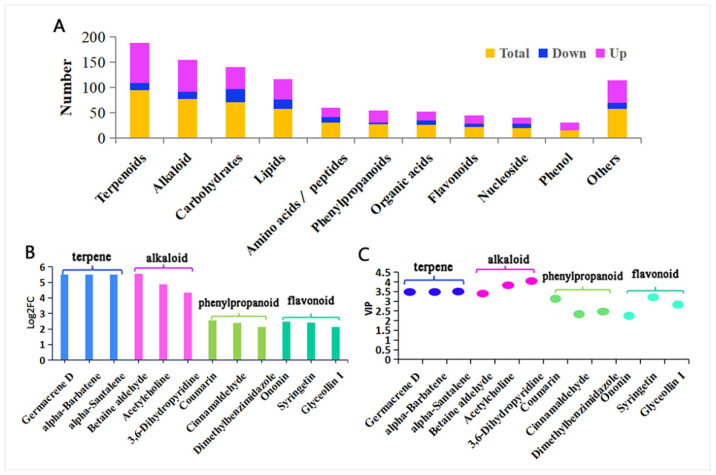
Differentially expressed metabolites in cotton. (**A**) Category of differentially expressed metabolites. (**B**) FC analysis of differentially expressed secondary metabolites related to cotton defense. (**C**) VIP analysis of differentially expressed secondary metabolites related to cotton defense.

**Figure 2 insects-15-00254-f002:**
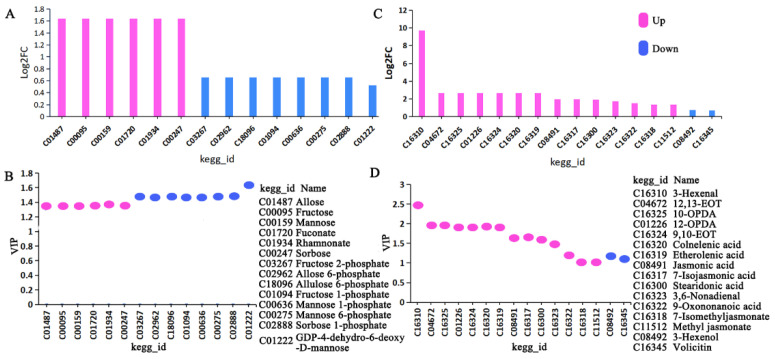
FC and VIP analysis of metabolism pathway. (**A**) FC and (**B**) VIP of fructose and mannose metabolism pathway. (**C**) FC and (**D**) VIP of ALA metabolism pathway.

**Figure 3 insects-15-00254-f003:**
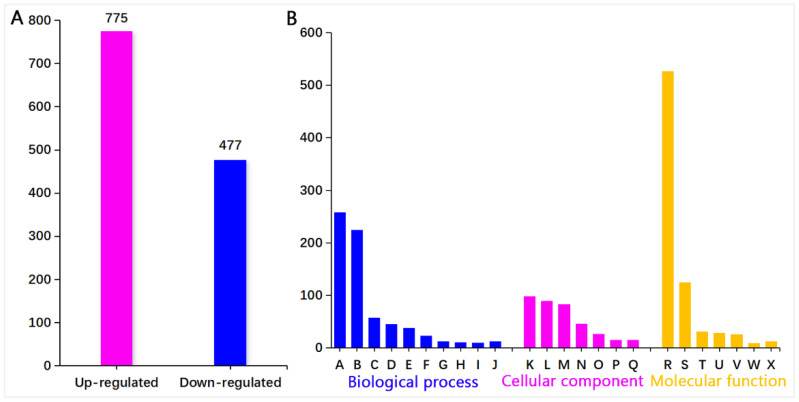
Protein identification and GO enrichment analysis of DEPs. (**A**) Up- and downregulated protein in cotton plants after mirid bug feeding. (**B**) GO annotation and functional classification of differentially expressed proteins in cotton plants, categorized as follows: biological process (A: metabolic process, B: single-organism process, C: cellular process metabolic process, D: response to stimulus, E: localization, F: reproductive process, G: signaling, H: cellular process, I: cellular component organization or biogenesis, J: others); cellular component (K: organelle part, L: organelle, M: cell part, N: membrane part, O: macromolecular complex, P: extracellular region, Q: membrane); and molecular function (R: catalytic activity, S: binding, T: transporter activity, U: structural molecule activity, V: antioxidant activity, W: enzyme regulator activity, X: others).

**Figure 4 insects-15-00254-f004:**
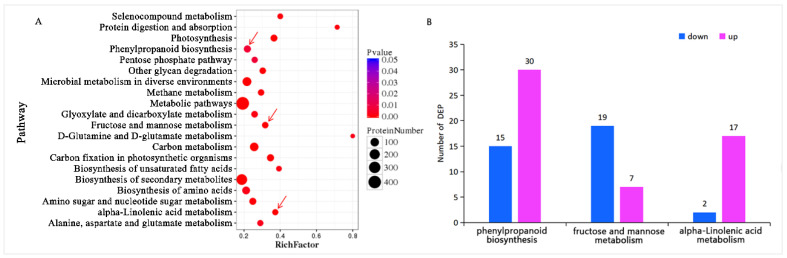
KEGG pathway enrichment. (**A**) Top 20 enriched pathways. Arrows represent metabolic pathways of interest. (**B**) Different metabolite changes of the defense-related metabolic pathways.

**Figure 5 insects-15-00254-f005:**
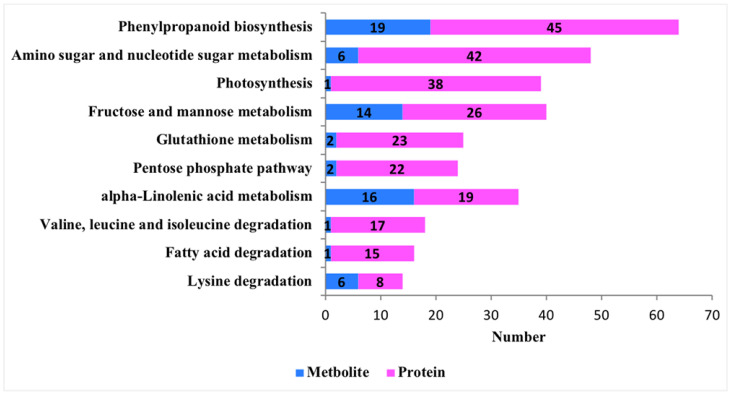
Common metabolic pathways of differential proteins and differential metabolites. Pink represents the number of differentially expressed metabolites (DEMs); blue represents the number of differentially expressed proteins (DEPs).

**Figure 6 insects-15-00254-f006:**
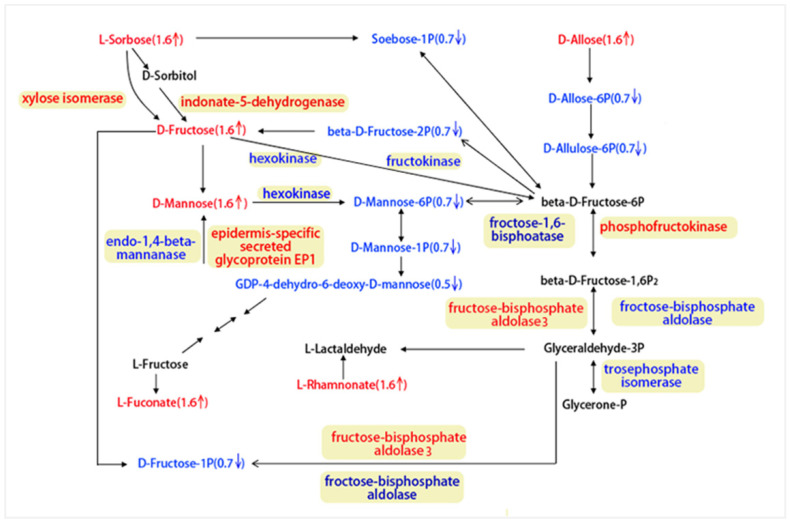
Integrated metabolomic and proteomic analysis of the fructose and mannose metabolism pathway. Red represents upregulated protease or metabolites, blue represents downregulated protease or metabolites, and black represents no change happening. The upregulated proteases were xylose isomerase (CotAD_24396, CotAD_63919, CotAD_02520), idonate-5-dehydrogenase (Cotton_D_gene_10030706), epidermis-specific secreted glycoprotein EP1 (CotAD_00556), phosphofructokinase (Cotton_D_gene_10039158), and fructose-bisphosphate aldolase3 (CotAD_58316). The downregulated proteases comprised hexokinase (CotAD_39434), fructokinase (CotAD_45862, CotAD_26511, CotAD_36008, CotAD_69191, Cotton_D_gene_10035332), endo-1,4-beta-mannanase (CotAD_57668), fructokinase-1-6-bisphosphatase(CotAD_39653), fructose-bisphosphate aldolase (CotAD_24238, CotAD_30698, CotAD_30700, CotAD_58944, CotAD_27060, CotAD_24236, CotAD_58035, CotAD_52093) and triosephosphate isomerase (CotAD_15944, CotAD_23341). The numbers in brackets represent FCs.

**Figure 7 insects-15-00254-f007:**
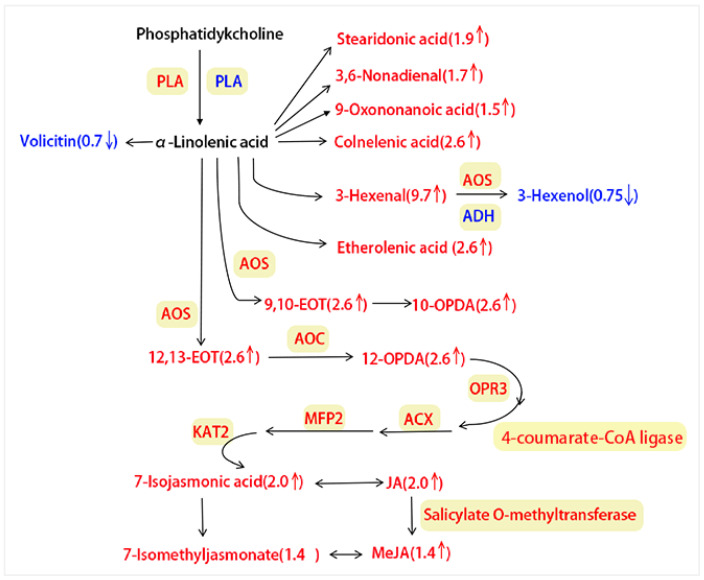
Integrated metabolomic and proteomic analysis of the ALA metabolism pathway. Red represents upregulated protease or metabolites, blue represents downregulated proteases or metabolites, and black represents no change happening. The upregulated proteases were: PLA (CotAD_52791), AOS (CotAD_35840, Cotton_D_gene_10023640, CotAD_58616), AOC (Cotton_D_gene_10007846, Cotton_D_gene_10007844), OPR3 (CotAD_59461, Cotton_D_gene_10037325), 4-coumarate-CoAligase (CotAD_59374), ACX (CotAD_52391, CotAD_12782, Cotton_D_gene_10040584), MFP2 (CotAD_18083), KAT2 (CotAD_51900, CotAD_10744, CotAD_66777), salicylate O-methyltransferase (CotAD_27039). The downregulated proteases comprised PLA (CotAD_64088) and ADH (CotAD_64304). The numbers in brackets represent FCs.

**Figure 8 insects-15-00254-f008:**
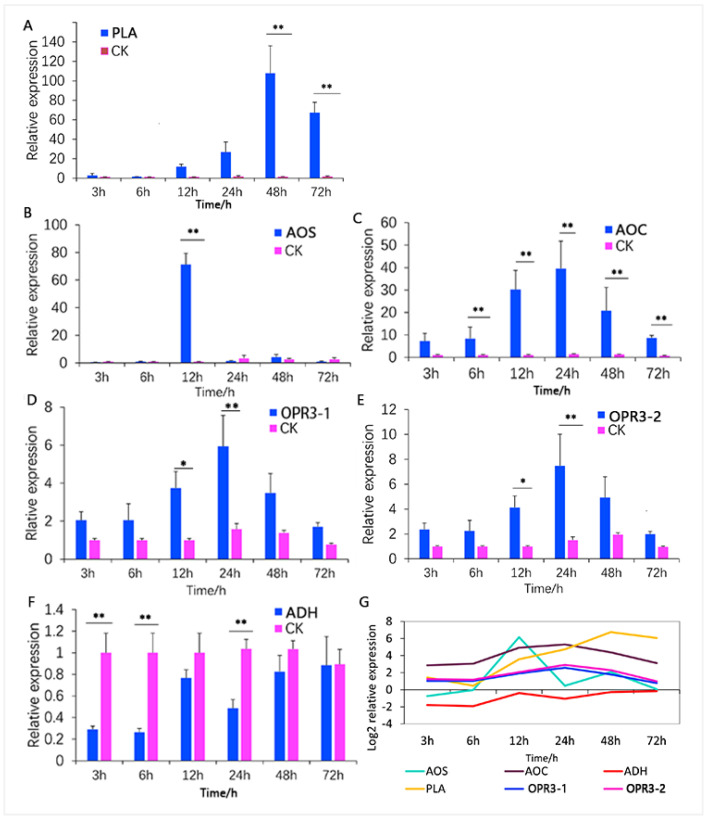
qRT-PCR analysis of the key genes in ALA metabolism pathway in cotton leaves. (**A**) Key regulator gene *PLA*, (**B**) key regulator gene *AOS*, (**C**) key regulator gene *AOC*, (**D**) key regulator gene *OPR3-1*, (**E**) key regulator gene *OPR3-2*, (**F**) downregulated gene *ADH*, (**G**) dynamic variation in the six ALA metabolism pathway-related genes in cotton plant. “*” represents *p* < 0.05 and “**” represents *p* < 0.01.

**Figure 9 insects-15-00254-f009:**
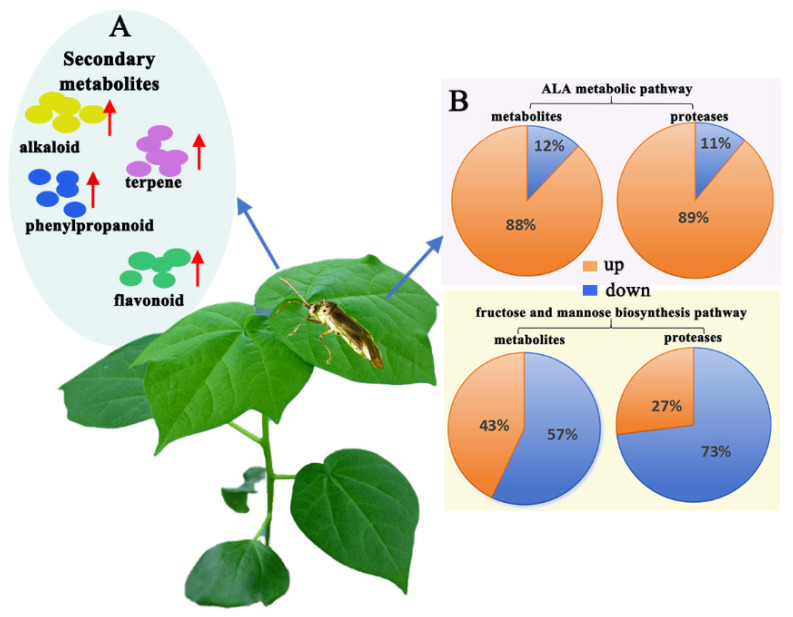
Hypothetic model of *A. suturalis*-induced cotton plant resistance. (**A**) Changes in secondary metabolites; (**B**) two key metabolic pathways.

**Table 1 insects-15-00254-t001:** Top 15 KEGG pathway analysis of differential metabolites of cotton induced by *A. suturalis*.

No.	Pathway_ID	KEGG Pathway Name	Total	Down	Up
1	map00909	Sesquiterpenoid and triterpenoid biosynthesis	40	0	40
2	map00590	Arachidonic acid metabolism	28	7	21
3	map00380	Tryptophan metabolism	23	5	18
4	map00904	Diterpenoid biosynthesis	20	6	14
5	map00950	Isoquinoline alkaloid biosynthesis	19	5	14
6	map00960	Tropane, piperidine and pyridine alkaloid biosynthesis	19	2	17
7	map00940	Phenylpropanoid biosynthesis	18	2	16
8	map00350	Tyrosine metabolism	17	2	15
9	map00592	alpha-Linolenic acid metabolism	16	2	14
10	map00902	Monoterpenoid biosynthesis	15	5	10
11	map00051	Fructose and mannose metabolism	14	8	6
12	map00360	Phenylalanine metabolism	14	2	12
13	map00903	Limonene and pinene degradation	14	3	9
14	map00130	Ubiquinone and other terpenoid-quinone biosynthesis	13	3	10
15	map00330	Arginine and proline metabolism	11	2	9

## Data Availability

It may be found in the online version of this article. All data generated and analyzed during the current study are included in this article.

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
