# Peer review of "Integrated Omics Analysis Reveals Key Pathways in Cotton Defense against Mirid Bug (*Adelphocoris suturalis* Jakovlev) Feeding"

_insects, 2024, doi:10.3390/insects15040254_

Round 1

Reviewer 1 Report

Comments and Suggestions for Authors

This manuscript did a lot of work to test the variations in metabolites and proteins of the plant cotton before and after insect feeding to try to understand the defense mechanisms developed by plant in response to insect feeding. I think this manuscript can provide important information for better understanding the defense mechanisms for plants against herbivores. However, this manuscript should be improved  before it can be accepted for publication. Data especially about figures were not organized well and are too redundant. Authors should trim and only show the most important findings. Below are my comments.

1. A key point about this research in cotton defense mechanism after insect feeding is how those changes in metabolites and proteins are trigged. Do authors have any comments about it? Can authors provide some information in the discussion part? Understanding this trigger mechanism will be possibly important to be employed for pest management.

2. Too many figures up to 16 figures showed, it hurts the most important points of this manuscript. Too many figures make authors feel somehow unfocused when reading though this manuscript. Authors should trim to only show the most important findings. It is not necessary to show every result as long as it does not hurt the main findings. 

3. Figure7-8, authors used these two figures to changes in two different pathways. But I think it is too complicated and too much information is showed on the figure. Authors should edit to make it as concise as possible for better showing the important findings. 

4. Authors used figures 9-15 to show their analysis of key regulators in ALA metabolism pathways. Every regulator was shown individually in separate figure.  Each figure has data from 4 different parts of cotton. I think it is sufficient to demonstrate their findings with only showing the leaf part because the experimental insects were put on the leaf and also I think insects mainly feed on the leaf during the experiment. As thus, all these results of key regulators can be combined into one figure make the result more straightforward and concise. 

5. In the manuscript, authors  actually focused two different pathways, ALA metabolism pathway and fructose and mannose metabolism pathway. But they only tested the key regulators in the former pathway, why the other pathway is not tested in terms of key regulators? 

6. As a model, figure 16 is also too complicated in terms of the findings in this manuscript. It will be not necessary be more helpful with as much as information shown in the model. For example, what is the purpose of panel C? 

7. Line 74, the sentence needs edition because it is confused to understand the logic. 

8. In terms of Figure 7, some contents seem from copy and paste, for example L-Fucose. Authors should revise to make it look better if it is thought to be necessary to show this Figure in the revised manuscript. 

Reviewer 2 Report

Comments and Suggestions for Authors

The manuscript Integrated "Omics Analysis Reveals Key Pathways in Cotton Defense Against the Mirid Bug (Adelphocoris suturalis Jakovlev) Feeding" by Liu et al. explores the upregulation of cotton metabolites as a stress response to feeding by A. suturalis. The results are valuable and the methods chosen by the authors are very interesting and state-of-the-art. However, the manuscript has some weak spots. The introduction part is very limited and contains little to no details about the issue to be studied. The discussion faces the same problem: it is weak, doesn't analyse the findings and doesn't compare them to previous studies.

I suggest the authors re-write those parts and make them detailed and supported by references.

Please find attached some more specific comments.

Comments on the Quality of English Language

The introduction is well-written but the rest of the manuscript needs proofreading. Language used in introduciton (bigger, more complete and thorough sentences) does not match language of the rest of the manuscript (short sentences wrongly syntaxed in most cases).

Reviewer 3 Report

Comments and Suggestions for Authors

The manuscript entitled "Integrated Omics Analysis Reveals Key Pathways in Cotton Defense Against the Mirid Bug (Adelphocoris suturalis Jakovlev)" by Hui Lu et al. elucidates specific alterations in the metabolism and protein expression of cotton plants triggered by A. suturalis feeding. These molecular responses offer insights into the intricate interaction between cotton plants and A. suturalis. The study's investigation was comprehensive and analytically robust. However, there are several issues that require attention prior to publication, outlined as follows:

 1. In the abstract section, it would be beneficial to include percentages for the identified "17 proteases" and "14 metabolites," as this would align with the information presented in Figure 16.

 2. The introduction section could benefit from improved organization, particularly regarding lines 70-73. The transition from "These reports identified..." to "desirable objective for cotton production" could be smoother.

3. Statements such as "Plants have evolved intricate mechanisms to respond to herbivore attacks, reshaping transcriptomes, proteomes, and metabolomes upon detecting physical and chemical cues from herbivores" (line 489) and "Currently, most biological control mechanisms against insect crop pests focus on predatory insects" (line 494) should be supported by relevant literature references.

 4. The discussion section should include broader implications for cotton breeding programs to provide a more comprehensive analysis of the findings.

 5. Please ensure uniform formatting of the references and review them carefully for accuracy and consistency.

Comments on the Quality of English Language

The English writing level of the manuscript shows a basic adherence to the requirements for publication. The content generally meets the standards expected for submission, with clear organization and coherent argumentation throughout. However, there are areas where further refinement and modification of language are needed to enhance clarity and precision. Certain sentences may benefit from restructuring or simplification to improve readability and ensure that the intended message is effectively conveyed. Additionally, attention should be paid to grammar, spelling, and punctuation to ensure accuracy and professionalism. Overall, while the manuscript demonstrates potential for publication, additional revisions are necessary to elevate the quality of the writing and meet the standards of academic publishing.

Round 2

Reviewer 1 Report

Comments and Suggestions for Authors

This manuscript has been much improved. I think it has good quality now and can be published. I have no more questions. 

Author Response

We appreciate for your comments,and thank you so much for your time.

Reviewer 2 Report

Comments and Suggestions for Authors

Dear authors,

thank you for taking the reviewers' comments into account and substantially improving your manuscript.

The introduction and discussion of the manuscript contain more information and details as requested.

Comments on the Quality of English Language

I suggest a native English speaker revise the manuscript, because there are still some grammar inconsistencies/contextual errors to be improved.

Author Response

We appreciate for your comments,and thank you so much for your time.

And we have examined the manuscript very carefully, and found some grammar and spelling mistakes, and revised all of them in green.

Reviewer 3 Report

Comments and Suggestions for Authors

No further comments. 

Author Response

(The authors gave the same response as above.)
